# Core Tokensets for Data-efficient Sequential Training of Transformers

## Abstract

Deep networks are frequently tuned to novel tasks and continue learning from ongoing data streams. Such sequential training requires consolidation of new and past information, a challenge predominantly addressed by retaining the most important data points - formally known as coresets. Traditionally, these coresets consist of entire samples, such as images or sentences. However, recent transformer architectures operate on tokens, leading to the famous assertion that an image is worth 16x16 words. Intuitively, not all of these tokens are equally informative or memorable. Going beyond coresets, we thus propose to construct a deeper-level data summary on the level of tokens. Our respectively named core tokensets both select the most informative data points and leverage feature attribution to store only their most relevant features. We demonstrate that core tokensets yield significant performance retention in incremental image classification, open-ended visual question answering, and continual image captioning with significantly reduced memory. In fact, we empirically find that a core tokenset of 1% of the data performs comparably to at least a twice as large and up to 10 times larger coreset.

## 1 Introduction

Deep learning models are rarely deployed in static environments and instead need to be continuously adjusted to their ever-changing surroundings. As distribution shifts occur over time, we require models to retain their performance on previous settings while accommodating new examples (Hadsell et al., 2020; Kudithipudi et al., 2022; Mundt et al., 2023). Naively, re-training a model from scratch for every change in the environment quickly becomes unfeasible for complex architectures with billions of parameters (Touvron et al., 2023; Brown et al., 2020). Consequently, research has focused on identifying small, representative subsets of the training data (Lopez-Paz & Ranzato, 2017), which can later be replayed continuously to avoid (catastrophic) forgetting (McCloskey & Cohen, 1989) of previously acquired information (Graham et al., 2021; Hassani et al., 2021; Xu et al., 2021). Formally, these (weighted) subsets can be described as *coresets*; data summaries that approximate the original loss function with a negligible loss in performance (Bachem et al., 2015; Blömer et al., 2016; Trapp et al., 2022). However, whereas multiple successful coreset selection techniques were proposed (Mirzasoleiman et al., 2020; Killamsetty et al., 2021a), we argue that modern architecture advances now allow for a yet to be leveraged level of data summarization. In particular, (vision) transformers now conveniently define each data input as an ordered set of tokens Vaswani et al. (2017); Dosovitskiy et al. (2020). In turn, we posit that it is not only a subset of the data instances that allows effective summarization but also only a handful of features within every data point that may be relevant to generalizing the task at hand. We refer to the latter as *core tokens* and, respectively, the subset of most meaningful tokens of important data instances as *core tokensets*.

Core tokensets build on the general principle of coresets, where we aim to identify the tokens that approximate the loss within a negligible error margin of the full dataset. To this end, we demonstrate that the attribution maps calculated using the attention scores across the transformer layers, as a means to understand the relevance of each token, can serve as the basis for core token selection. In practice, we thus explore different strategies to determine the token's relevance, e.g. by exploiting the gradients or perturbing the attention scores. Our first interesting insight is that naively retaining only core tokens from *each* data sample at a specified rate results in a similar cost as an involved coreset (that selects a subset of data instances) with an identical retention rate. However, and more importantly, we can couple techniques in a two-fold subset selection approach, i.e., our so-called

Figure 1: Continued training with core tokensets. In sequential training, forgetting previous tasks is mitigated by selecting the most meaningful previously observed data points and storing only their most influential tokens (yellow shaded parts; left side). The resulting core tokenset is then interleaved in the training of new tasks (blue shaded parts; right side), where random input dropout is further employed to render the model susceptible to observing partial inputs.

core tokensets, where we first identify a subset of samples using coreset selection methods and then select core tokens of those samples. A schematic overview of our framework is depicted in Fig. 1. We demonstrate the benefits of core tokensets in the scope of sequential training tasks, where we show that storing only the relevant partial inputs yields informative summaries with both improved performance as well as significantly reduced memory buffer size in comparison to state-of-the-art traditional core sets. In summary, we make the following contributions:

- We introduce the notion of core tokensets, a sub-data summary that selects a subset of most important tokens for a subset of most important data instances of a dataset.
- We show how to leverage the attribution maps computed across the transformer layers in an exploration of various strategies to select the most influential tokes.
- Finally, we extensively highlight the empirical performance and memory benefits of core tokensets in different sequential task setups in image classification, multi-modal image captioning, and multi-modal visual question-answering (VQA).

## 2 PRELIMINARIES AND RELATED WORK

**Coresets:** Conventional dataset summaries are constructed as instance-wise (weighted) subsets of the entire training data. These coresets are comprised of the most informative instances with an aim to approximate the full dataset with a negligible difference in the cost function. That is, we wish to find the subset $C_x$ of $D$, assuming an additively, into non-negative functions decomposable, cost function for our given dataset $D$ (Bachem et al., 2015; 2017), such that for any solution (parameters) $Q$:

$$|\texttt{cost}(D, Q) - \texttt{cost}(C_x, Q)| \leq \epsilon \cdot \texttt{cost}(D, Q), \quad where \; \epsilon > 0 \tag{1}$$

In practice, coresets were primarily designed to demonstrate dataset approximation capabilities for traditional machine learning models such as K-means(Blömer et al., 2016; Feldman et al., 2007; Har-Peled & Kushal, 2005), SVM (Dong et al., 2005), and logistic regression (Huggins et al., 2016), with very few works aimed at deep neural networks (DNNs). For the latter, Killamsetty et al. (2021b) resorts to bi-level optimization with coresets selection as an outer objective and the generalization of model parameters as an inner objective. However, it requires several optimization steps, which can be impractical for complex architectures. As an alternative to bi-level optimization, gradient-based coreset approaches have thus emerged with an aim to solve an alternative to Eq. 1 that operates on the gradients of the loss rather than approximating the loss directly. CRAIG (Mirzasoleiman et al., 2020) thereby, aims to identify the optimal coreset by converting the gradient matching problem to a

monotone submodular function optimization problem and solving it with a pre-defined error bound. Similarly, GradMatch (Killamsetty et al., 2021a) employs full dataset gradients while additionally employing L2 regularization to reduce dependency on any specific data sample.

In sequential learning, coresets can effectively be used as replay buffers to interleave in continual training (Mundt et al., 2023) to avoid catastrophic forgetting (McCloskey & Cohen, 1989). However, prominent replay strategies rely on ad-hoc subset selection, such as reservoir or random sampling (Rolnick et al., 2019; Kumari et al., 2022; Tiwari et al., 2022), to retain previously acquired information over the course of training. In our work, we evaluate the impact on transformers to solve more complicated tasks beyond standard classification problems, such as sequential VQA & image captioning and propose an advanced data summarization technique, referred to as *core tokensets*.

**Transformer preliminaries:** Given a sample $X$, transformers (Vaswani et al., 2017) make use of a sequence of tokens $x_p$, where each token captures distinctive features of the data sample. Usually, for images, the samples are transformed into $T$ equally sized $P \times P$ flattened 2D patches $x_p \epsilon \mathbf{R^{T \times (P^2 F)}}$ ($F$: number of feature channels). Subsequently, each patch is encoded into a single token embedding $\mathbf{E}^t \in \mathbf{E}$; every embedded patch is then concatenated with its corresponding positional embedding vector $\mathbf{E}_{pos}^t \in \mathbf{E}_{pos}$. The self-attention layers then process these sequences of token embeddings $z_0$:

$$z_0 = [x_{cls}; x_p^1; x_p^2; ...; x_p^T]\mathbf{E} + \mathbf{E}_{pos} \tag{2}$$

## 3 CORE TOKENSETS: SUMMARIZING DATASETS WITH INFORMATIVE TOKENS

As transformers are now capable of processing data inputs as a sequence of multiple tokens, we now present a novel way to summarize datasets even more efficiently than traditional coresets.

**Definition 3.1** (Core tokensets). A core tokenset $C_t$ is a subset of tokens ($\subseteq T$) of a dataset $D$, such that a solution $Q$ found on the core tokenset approximates the cost of the full dataset within negligible error margin: $|\text{cost}(T, Q) - \text{cost}(C_t, Q)| \leq \epsilon \cdot \text{cost}(T, Q)$.

This definition is a modification of core set equation 1. Instead of being based on data instances, it is now based on tokens and we can thus leverage the unique architectural properties of modern transformers to process inputs at a sub-data point level. In order to identify the *core tokens*, we associate each input token $x_p^t$ with a binary importance variable $v^t$:

$$C_t := \{(x_p^t, t)|v^t = 1, 0 <= t < |z_0|\} \ s.t.|C_t| < T \tag{3}$$

Note that each stored core token remains associated with its position $t$, preserved in the form of integer indexes of the extracted tokens. This will later be used to preserve the original structure in continued training. The importance variable will typically be regulated by the relevance of the token to its current task, but we can also think of straightforward selection methods. For instance, a naive (random) selection can be realized by sampling the importance variable from a Bernoulli distribution: $v^t \sim Bernoulli(r)$. Here, $r$ naturally maps to our key hyperparameter, i.e. the retention rate that reflects the percentage of tokens retained from $T$.

Whereas there exist well-defined strategies to determine coresets, methods to select *core tokens* are yet to be explored. Even though random sampling is a feasible baseline in forming memory Hayes et al. (2021), randomly selecting core tokens can easily lead to storing redundant tokens. In particular, in images, the object of interest often only occupies tiny regions of an input sample; storing the background alone is unlikely to approximate the true loss well. As a remedy, we propose leveraging attribution techniques and investigating the attention maps for token relevance to construct our set of core tokens. First, each token is assigned a relevance score, determining its contribution towards the prediction. We then draw inspiration from "explainability" techniques (XAI) (Chefer et al., 2021b; Deiseroth et al., 2023) to elevate the efficacy of attribution maps further to calculate the token relevance score. For instance, we can perturb the attention scores across the blocks or calculate the gradients across the transformer blocks to quantify the contribution towards the final prediction.

Let us specifically outline the gradient-based approach to select core tokens. Given a transformer model with B blocks, we first calculate the attention map ($A^b$) for each block ($b \in B$) comprising h attention heads. We consider the Hadamard product of the gradients of the attention map ($\nabla A^b$)

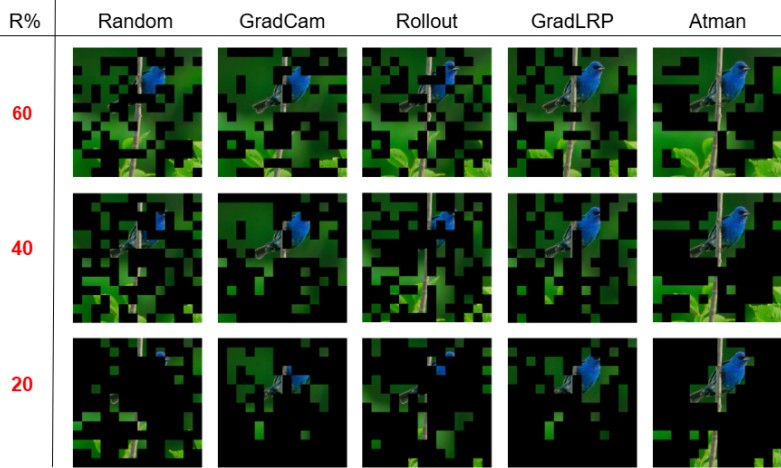

Figure 2: Visualization of the token relevance map constructed with the help of different core token strategies as we gradually reduce the retention rates (R) for the tokens to be retained from a sample

and the layer-wise relevance score ($L_R$) (Bach et al., 2015) with respect to a target class. The final attribution map $\bar{A}^b$ is achieved by calculating the mean ($E_h$) of the product across h attention heads in addition to an identity matrix to account for skip connections in transformers. The final influence matrix $S_{T \times T}$ is then defined by multiplying all the attention head scores, where each row consists of the relevancy score of each token compared to the other token.

$$\bar{A}^b = I + E_h(\nabla A^b \odot L_R); \quad S_{(T \times T)} = \bar{A}^{(1)} \cdot \bar{A}^{(2)} \cdot \bar{A}^{(3)} \cdot \bar{A}^{(B)} \tag{4}$$

From the influence matrix, we extract the relevance of each token, which will ultimately determine the individual binarized $v_t$ (see Fig. 2). For classification, we only consider the class token in the first row. The top values then lead to extraction of the core token into our set. The cardinality of the set is pre-determined based on the chosen retention rate $R$ such that $\frac{|C_t|}{|T|} = R$. Intuitively, such a gradient-based core token selection strategy is related to earlier described gradient matching strategies for core set selection. We determine the most influential tokens of each data point via analysis of the gradients and store a subset that approximately leads to the same outcomes as the entirety of tokens.

## 4 SEQUENTIAL TRAINING WITH CORE TOKENSETS

When we train our transformer sequentially over time, we optimize our model simultaneously on newly arriving data of the novel task and the memory buffer data as modeled through the *core tokenset*. The training loss $\mathcal{L}$ is thus:

$$\mathcal{L} = \mathbb{E}_{(x,y)} \mathcal{L}(f(x),y) + \mathbb{E}_{(x_p^c,y) \in C_t} \mathcal{L}(f(x_p^c),y) \tag{5}$$

where $\mathcal{L}(.,.)$ denotes an arbitrary loss (e.g. the cross-entropy), $f$ denotes the model predictions $f = z(g(x))$ and $g(x)$ is the shared encoder model. Similarly, $f(x_p^c)$ denotes the predictions made on the core tokenset memory buffer. Before we proceed to experimentally corroborate our core tokensets, let us first consider two further imperative aspects: 1.) how to make transformers susceptible to the partial inputs (subset of tokens) stored in our data summary, and 2.) provide an intuition for the trade-offs and interplay between storing a traditional core set and a subset of tokens.

### 4.1 MAKING TRANSFORMERS SUSCEPTIBLE TO PARTIAL INPUTS

During regular (pre-)training, ViTs only observe complete inputs of $T$ tokens. If we now feed them with a subset of tokens, i.e. drop a subset of the concatenated patches, this results in an input sequence that strongly deviates from the expected input structure. This, unfortunately, immediately leads to unexpected model behavior and performance degradation. In fact, in Fig. 3a we report the latter, in an experiment where we transition from training on complete samples during initial training to

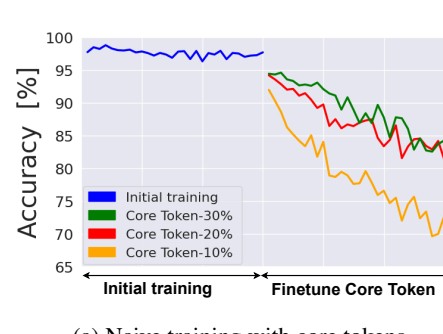 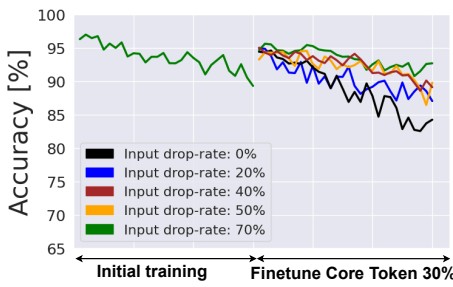

(a) Naive training with core tokens          (b) Training with random dropping of input tokens

Figure 3: Transformers are not susceptible to partial inputs. We illustrate the fine-tuning behavior of a ViT-B/16 model on 20 random classes of ImageNet using core tokens. (a): Without countermeasures, performance drops massively as the model is unable to handle partial inputs. (b) Randomly zeroing-out tokens (through input dropout) of complete data points during training makes the model amenable to learn from core tokens.

fine-tuning on partial inputs with *core tokensets*. Specifically, a pre-trained ImageNet (Deng et al., 2009) Vit-B/16 model is continuously fine-tuned to 20 random classes first on all data and then on *core tokenset* with different retention. For simplicity, we randomly select the core tokens. We can observe a considerable drop in performance of roughly 15% in prediction accuracy at 30% retention.

If one were to draw premature conclusions, one would posit that core tokens are ineffective. However, this is not the case and our initial deteriorated model performance is truly due to the model encountering unexpected input. As a remedy, we demonstrate that we need only make the model amenable to observing partial inputs. Whereas we could conceive several notions to achieve the latter, an effective simple strategy is to ensure that the model is always prompted with a complete sequence of $T$ tokens for core tokens. To that extent, the input value of left out tokens is set to **0**. Crucially, this setup preserves the input structure on which the model was pre-trained, thus ensuring that spatial relations of patches are represented correctly. To emulate the respective sparsity of core tokensets, we additionally randomly zero-out tokens of full data samples according to the expected retention rate, similar to dropout (Srivastava et al., 2014) applied to the input. In Fig. 3b, we again tune a pre-trained vision transformer (ViT) on 20 classes of ImageNet but now with random dropping and subsequently fine-tuning on sets of core tokens. We fix the retention rate $R$ at 30% for *core tokens* while varying the drop rate used during the initial 20-class training. We see that the use of dropping tokens is imperative to avoid performance drops. As we match the drop rate with the retention rate, the model becomes more adept at learning from core tokens. Eventually, we retain the initial performance, much in contrast to the earlier curve (also green, in panel a) without any training amendments.

For sequential learning on the combined token $[\{x_p^t * v^t\}_{t=1}^T \uplus \{x_p^c\}_{c=1}^{C_t}]$, we thus adopt above practical strategy. That is, we additionally randomly sample $v^t \sim Bernoulli(r)$ for the newly observed complete data repeatedly during training, i.e. we randomly drop input tokens. The core tokenset already has been attributed its relevance, i.e. a respective $v^c$ is always unity following one of the earlier described selection strategies. It is here, where we make use of the stored integer positions of the core tokenset in order to pad the remainder with zeros. In Fig. 4, where we sequentially tune on ImageNet subsets sequentially, our strategy consistently retains performance, whereas naive training on the core tokenset deteriorates heavily.

## 4.2    Intuition on Effectiveness: Coresets vs. Core Tokensets

Our definition 3.1 of core tokens naturally raises the question of whether it is more beneficial to store subsets of tokens or subsets of data points, or whether to combine the latter. To provide intuition for this question, we conceive two practical variants of data summarization. The first strives to identify the most relevant subset of tokens for every data point as a data summary which we will refer to as only extracting *core tokens*. Secondly, we would instead attempt to find the most important subset of data points and respectively their most influential tokens. We will refer to this latter when we speak of *core tokensets* in the remainder of the paper's experiments. In other words, core tokenset follow

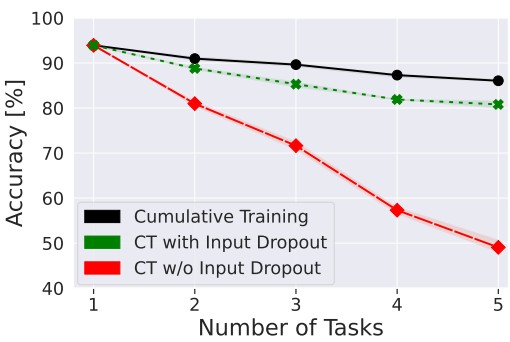 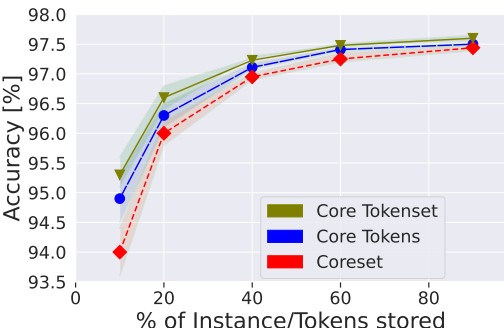

Figure 4: Influence of token dropout. We perform continual training (CT) on five distinct sub-tasks. Randomly dropping tokens of novel inputs leads to an improvement of 40% in accuracy, in contrast to a scenario where the model is unable to handle the partial input of core tokensets.

Figure 5: Comparing core instances with core tokens. The accuracy of a ViT model on a CT task is shown. A memory buffer with core tokens outperforms traditional core sets across different memory sizes (x-axis), whereas their conjunction into core tokensets provides even more benefit.

earlier Fig. 1, where we first build a coreset and then subsequently identify the core tokens for the selected samples, whereas only selecting core tokens acts as an ablation omitting the coreset step.

In Fig. 5 we evaluate these two strategies, only core tokens and core tokensets, against traditional core sets for our pre-trained ViT-B model on 20 random ImageNet classes with varying retention rates (10-100%). We can observe that for all retention rates, training on core tokens outperforms traditional core sets. However, core tokensets, identifying the most important tokens for the most important data points, yield substantial further performance improvement. We believe this to be fairly intuitive. For instance, consider a budget where we store only 1% of the samples in our memory buffer; coreset techniques, most evidently, will throw away 99% of the total data instances. Meanwhile, for core tokens, it will be possible to store 1% of information for every single item of dataset. In contrast, for core tokensets, the combination allows us to select more instances than the coreset technique by focusing on a few core tokens per sample. To be precise, we can store 10% of the samples ($> 1\%$ coresets) and 10% of the tokens ($> 1\%$ core tokens) of those samples at the same memory budget. We continue to extensively corroborate these findings in our experimental section.

## 5 EXPERIMENTS

We now extensively evaluate core tokensets on three diverse sequential tasks: 1) image classification, 2) open-ended visual question answering, and 3) image captioning. Specifically, we want to quantify a) the model's stability due to the informativeness of the memory in a comparison of various subset selection strategies b) the relation of obtained performance to the overall cost of the memory buffer.

### 5.1 EXPERIMENTAL DETAILS

**Datasets and Models.** We use ImageNet100 (Deng et al., 2009) for classification, which we sequentially segment into five tasks with 20 unique classes each. For VQA, we train on three separate datasets in sequence, namely VQA 2.0 (Antol et al., 2015), CLEVR (Johnson et al., 2017), and Visual Genome (Krishna et al., 2017). For the image captioning experiments, we rely on MS COCO (Lin et al., 2014), which we segment into five equally sized tasks based on the object classes. We start with a pre-trained VIT-B/16 trained on ImageNet21k as our base model (Wightman, 2019) for image classification to focus our investigation on data summary efficiency and knowledge retention. Lastly, we use a pre-trained BLIP model (Li et al., 2022) as initialization for both VQA and image captioning tasks. More details for datasets and training set-ups are provided in Appendix A.2. Depending on the complexity of the task, we have trained our models on 8-12 Nvidia A100 GPUs.

**Strategies.** We evaluate two recent state-of-the-art coreset approaches, namely CRAIG (Mirzasoleiman et al., 2020) and GradMatch (Killamsetty et al., 2021a), both relying on gradients for their

Table 1: Obtained final accuracy in sequential image classification for different subset selection strategies as a function of retained data percentage. Extraction of core tokens performs comparably to modern core set selection methods. However, the conjunction into core tokensets, here specifically combining GradMatch and Atman, consistently yields large accuracy improvements. Color coding is further used to highlight similar performance of a traditional state-of-the-art coreset and our proposed core tokensets. Across various retention rates, core tokensets can even be observed to perform comparably or favorably to much larger sized alternative approaches.

| | Methods | 80% | 60% | 40% | 20% | 10% | 1% |
|---|---|---|---|---|---|---|---|
| | | | | Retention rates: Sequential Image Classification | | | |
| Coreset | Random | $81.91_{\pm 0.09}$ | $78.19_{\pm 0.15}$ | $75.61_{\pm 0.24}$ | $71.21_{\pm 0.87}$ | $69.22_{\pm 0.43}$ | $67.21_{\pm 0.61}$ |
| | CRAIG | $82.97_{\pm 0.02}$ | $81.42_{\pm 0.02}$ | $79.34_{\pm 0.02}$ | $74.82_{\pm 0.05}$ | $73.99_{\pm 0.06}$ | $71.23_{\pm 0.08}$ |
| | GradMatch | $82.99_{\pm 0.02}$ | $81.58_{\pm 0.02}$ | $79.84_{\pm 0.02}$ | $74.92_{\pm 0.05}$ | $74.21_{\pm 0.08}$ | $71.67_{\pm 0.11}$ |
| Core Token | GradCam | $82.19_{\pm 0.09}$ | $78.69_{\pm 0.12}$ | $76.21_{\pm 0.01}$ | $71.33_{\pm 0.23}$ | $69.81_{\pm 0.31}$ | $67.81_{\pm 0.31}$ |
| | Rollout | $82.89_{\pm 0.09}$ | $79.98_{\pm 0.15}$ | $78.97_{\pm 0.04}$ | $72.23_{\pm 0.13}$ | $71.11_{\pm 0.23}$ | $69.24_{\pm 0.23}$ |
| | GradLRP | $83.43_{\pm 0.09}$ | $81.55_{\pm 0.09}$ | $80.03_{\pm 0.06}$ | $75.12_{\pm 0.21}$ | $74.23_{\pm 0.23}$ | $72.03_{\pm 0.23}$ |
| | Atman | $83.67_{\pm 0.05}$ | $82.11_{\pm 0.05}$ | $81.87_{\pm 0.07}$ | $76.98_{\pm 0.12}$ | $75.42_{\pm 0.12}$ | $72.43_{\pm 0.12}$ |
| | Core Tokenset | $84.11_{\pm 0.02}$ | $83.02_{\pm 0.03}$ | $82.74_{\pm 0.03}$ | $77.87_{\pm 0.06}$ | $76.63_{\pm 0.11}$ | $73.63_{\pm 0.11}$ |

data selection. In addition, we consider four distinct feature attribution methods to derive attention maps, based on which core tokens are selected. *Rollout* considers the information flow from the input layer to deeper ones (Abnar & Zuidema, 2022). In contrast, *Grad-Cam* relies on the gradients for each input token with respect to the ground-truth output. Going a step further, gradients with layer-wise relevance propagation (*Grad-LRP*) (Chefer et al., 2021b;a) considers the respective relevance of each layer individually. Finally, *AtMan* (Deiseroth et al., 2023) calculates the influence of each token by perturbing the attention scores. We provide detailed descriptions for each strategy in Appendix A.1.

**Evaluation Metrics.** In all incremental setups, we report the corresponding metrics averaged over all the tasks. We report averaged accuracy for image classification and VQA. However, note that we do not train VQA as a classification task. Instead, we open-endedly generate answers in natural language. Consequently, an answer is counted as correct if the generated text is the same as the ground-truth label. For image captioning, we report BLEU-4 (Papineni et al., 2002) and ROUGE (Lin, 2004), which measure the similarity between the model-generated text and the reference text.

## 5.2 SEQUENTIAL IMAGE CLASSIFICATION

In our sequential classification task, we specifically evaluate the informativeness of our memory buffer in preserving the old information by comparing two variants of coreset strategies with four ways of selecting core tokens. Tab. 1 compares the methods as a function of the stored proportion of the dataset. Here, we contrast coresets that select a subset of data points with pure extraction of core tokens that select a subset of tokens for all data points. These are then compared to core tokensets, which are based on the conjunction of the best-performing coreset and core token strategy. As a reference, the model achieves a top accuracy of *85.4%* when we sequentially train it on *all accumulated data* of every time step, i.e., data gets added into a growing dataset over time. We first notice a significant drop in performance with a gradual decline in retention rates when data is randomly selected. Here, the more sophisticated coreset selection techniques offer substantial improvement, with GradMatch being the slightly better approach. Turning to the specific approaches to identify influential

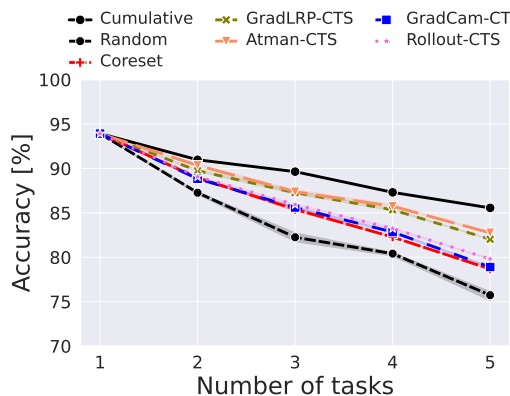

Figure 6: Comparison of core tokenset selection techniques for sequential image classification. Methods vary through feature attribution technique, yet all demonstrate improvements over a traditional GradMatch-based core set.

Table 2: Obtained final accuracy in incremental ImageNet-R, which we divided into 10 and 20 sequential tasks. We report results averaged over three trials. *ACC* denotes the achieved accuracy on the final task, whereas $\overline{ACC}$ represents the averaged accuracy over all tasks. Core tokensets are competitive with other approaches, even though we emphasize that they are complementary in nature.

| Task Splits: ImageNet-R(endition) | 10 | | 20 | |
|---|---|---|---|---|
| Method | $ACC_{10}$ (↑) | $\overline{ACC}_{10}$ (↑) | $ACC_{20}$ (↑) | $\overline{ACC}_{20}$ (↑) |
| *Joint* | $81.14 \pm 0.34$ | - | $81.14 \pm 0.34$ | - |
| *Sequential* | $46.07 \pm 1.15$ | $62.91 \pm 0.68$ | $34.62 \pm 0.85$ | $51.15 \pm 1.50$ |
| L2P Wang et al. (2022b) | $62.54 \pm 0.24$ | $67.98 \pm 0.27$ | $57.92 \pm 0.28$ | $64.57 \pm 0.29$ |
| DualPrompt Wang et al. (2022a) | $65.41 \pm 0.52$ | $69.39 \pm 0.43$ | $61.00 \pm 0.72$ | $65.80 \pm 0.67$ |
| CODA-P Smith et al. (2023b) | $71.47 \pm 0.35$ | $75.82 \pm 0.29$ | $67.28 \pm 0.30$ | $72.34 \pm 0.17$ |
| C-LoRA Smith et al. (2023a) | $71.89 \pm 0.45$ | $75.33 \pm 0.28$ | $65.71 \pm 0.60$ | $70.63 \pm 0.85$ |
| LAE Gao et al. (2023) | $71.70 \pm 0.39$ | $76.71 \pm 0.10$ | $66.98 \pm 0.35$ | $73.72 \pm 0.05$ |
| InfLoRA-b5 Liang & Li (2024) | $74.13 \pm 0.18$ | $78.54 \pm 0.14$ | $68.41 \pm 0.29$ | $74.00 \pm 0.50$ |
| InfLoRA Liang & Li (2024) | $75.65 \pm 0.14$ | $80.82 \pm 0.24$ | $71.01 \pm 0.45$ | $77.28 \pm 0.45$ |
| Core Tokensets | $\mathbf{75.95 \pm 0.15}$ | $\mathbf{81.32 \pm 0.14}$ | $\mathbf{71.58 \pm 0.22}$ | $\mathbf{77.59 \pm 0.25}$ |

tokens, perturbing attention scores, as in Atman,
stand out as the best-performing core token selection mechanism. GradLRP follows closely behind, whereas GradCam and Rollout fall short. We emphasize this difference through matching color coding, where we highlight the core tokenset performance that matches or even surpasses the corresponding GradMatch accuracy. The observable trend is that core tokensets yield similar performance at greatly reduced memory cost. This is most visible when 1 % of the data is retained, where the core tokenset performs similarly to a ten times larger core set. Both GradCam and GradLRP by themselves outperform the coreset strategies, reinforcing our findings of earlier Fig. 5. More importantly, our core tokenset, based on the conjunction of the best-performing GradMatch and Atman, yields improvements of several percentage points in accuracy over any of the other methods.

To demonstrate that this obtained advantage is consistent, we combine GradMatch with all four core token selection mechanisms to form core tokensets at a common dataset retention rate of 40 %. We show the respective evolution of the accuracy as tasks get learned sequentially in Fig. 6. Here, it can be seen that the Atman (yellow, triangle) and GradLRP (green, x) curves reasonably approach the upper bound on the full dataset. However, it is particularly noteworthy that when combined into a core tokenset with GradMatch, any core token selection mechanism improves performance over the traditional core set approach (red, dashed).

## 5.3 CONTEXTUALIZATION WITH SOTA INCREMENTAL LEARNERS ON IMAGENET-R

Before evaluating core tokensets on other domains, we empirically contextualize our proposed core tokenset approach within a wider body of current state-of-the-art incremental learning approaches. Importantly, this comparison is primarily intended to provide a larger context of core tokensets' capabilities. We argue not to draw simplified conclusions from these direct comparisons, since the respective methodologies differ vastly and depend on different assumptions. Rather, they should be interpreted as testimony to core tokensets efficacy and utility when prospectively being combined.

We chose ImageNet-R(endition) (Hendrycks et al., 2021) as a standard benchmark for challenging and diverse continual learning tasks. We report averaged accuracy across the sequential task of 10 and 20 data subsets in Tab. 2. Based on insights from the previous experiment, we built the core tokensets using GradMatch and AtMan at an 50% retention rate, as this choice provided a good trade-off between memory and efficacy. The results demonstrate that core tokensets remain competitive with best-performing methods like InfLoRA Liang & Li (2024).

## 5.4 OPEN-ENDED VISUAL QUESTIONING ANSWERING

We now move on to a larger-scale, multi-modal setup, where we finetune BLIP over 2 million incremental VQA pairs. In Tab. 3, we report results over the considered selection strategies at different retention rates. While GradMatch remains the best coreset selection strategy, GradLRP

Table 3: Obtained final accuracy in sequential VQA for different subset selection strategies as a function of retained data percentage. Table in analogy to Tab. 1. Color coding again highlights the best-performing coreset and core tokensets. Similar to image classification, core tokensets perform comparably or even better than a much larger sized core set in VQA tasks.

| | Methods | Retention rates: Visual Question Answering | | | | | |
| | | 70% | 50% | 30% | 20% | 10% | 1% |
|---|---|---|---|---|---|---|---|
| Coreset | Random | $77.02_{\pm0.11}$ | $74.86_{\pm0.07}$ | $71.23_{\pm0.04}$ | $69.88_{\pm0.25}$ | $67.23_{\pm0.33}$ | $66.03_{\pm0.34}$ |
| | CRAIG | $78.15_{\pm0.04}$ | $76.93_{\pm0.07}$ | $74.67_{\pm0.07}$ | $73.56_{\pm0.05}$ | $71.63_{\pm0.12}$ | $70.54_{\pm0.12}$ |
| | GradMatch | $78.17_{\pm0.04}$ | $77.03_{\pm0.04}$ | $75.03_{\pm0.03}$ | $73.93_{\pm0.06}$ | $71.77_{\pm0.10}$ | $70.62_{\pm0.11}$ |
| Core Token | Gradcam | $77.04_{\pm0.11}$ | $75.16_{\pm0.12}$ | $71.83_{\pm0.04}$ | $70.45_{\pm0.25}$ | $68.93_{\pm0.33}$ | $67.66_{\pm0.34}$ |
| | Rollout | $77.14_{\pm0.11}$ | $75.56_{\pm0.07}$ | $73.73_{\pm0.04}$ | $72.98_{\pm0.25}$ | $70.33_{\pm0.33}$ | $68.88_{\pm0.21}$ |
| | Atman | $77.16_{\pm0.05}$ | $75.93_{\pm0.05}$ | $74.15_{\pm0.04}$ | $73.29_{\pm0.11}$ | $71.05_{\pm0.06}$ | $70.46_{\pm0.11}$ |
| | GradLRP | $78.45_{\pm0.09}$ | $77.63_{\pm0.05}$ | $75.43_{\pm0.08}$ | $74.30_{\pm0.08}$ | $72.43_{\pm0.09}$ | $71.02_{\pm0.14}$ |
| | Core Tokenset | $\mathbf{78.70_{\pm0.03}}$ | $\mathbf{78.01_{\pm0.02}}$ | $\mathbf{75.99_{\pm0.02}}$ | $\mathbf{75.32_{\pm0.03}}$ | $\mathbf{73.35_{\pm0.05}}$ | $\mathbf{71.98_{\pm0.06}}$ |

now takes the lead over ATMAN for the identification of core tokens. Respectively, our VQA core tokenset combines GradMatch and GradLRP. Again, core tokensets consistently outperform all other methods across all retention rates. Similar to our image classification table, to offer better visualization, we have color-coded specific cells of coreset and core tokensets to highlight their comparable performances at contrasting retention rates. Again, core tokensets can be observed to perform much better or be significantly more memory efficient. For instance, should we wish to achieve a performance of at least 75 %, we would have to resort to a GradMatch-based core set that extracts a 30 % subset of the dataset. In contrast, core tokens already surpass our example performance criterion when retaining only 20% of the data. Once more, this effect gets amplified as we store less data, up to an astonishing performance at 1% stored data that outperforms a ten times larger core set. Consequently, the findings of our VQA experiments are consistent with those observed image classification.

## 5.5 Image Captioning

Finally, we conduct experiments on sequential image captioning tasks, comparing core tokensets (CTS) based on the various selection strategies at an aggressive retention rate of $10\%$. We have chosen this low retention rate as it already demonstrates results that are reasonably close to training on the full dataset, making it obsolete to investigate higher percentages. Instead, we report both BLEU-4 and ROUGE scores as a function of tasks over time in Fig. 7. Again, we report GradMatch as the favorable coreset selection strategy and combine it with different core token selection strategies. In full consistency with all prior findings, all CTS variants outperform the traditional coreset, whereas the GradLRP-based CTS takes the lead and is once more closely followed by Atman. The ordering of the methods is the same for both BLEU and ROUGE scores, with both demonstrating a marginal drop in performance when only 10% of the data is stored with a CTS.

## 6 Discussion

We have thoroughly evaluated core tokensets across three different sequential tasks. Despite the different nature of these tasks, our empirical findings provide consistent insights.

The first crucial insight lies in the efficacy when comparing retention of data subsets, i.e., traditional coresets, vs. core tokens, i.e., storing the relevant subset of tokens of each data point. If these methods were to be viewed as separate competitors, then it appears safer to reside on the token level. Our perhaps obvious hypothesis here is that intuitively, one loses less information when throwing away an informative token, in contrast to throwing away an entire data point. As an example, if the memory size is chosen to be, e.g., 10 %, it is more likely that a high amount of relevant information is still present in the top 10 % tokens of each data point, in contrast, to fully throwing away 90 % of the data instances (as is done in a traditional core set). This should be particularly true if the dataset features little redundancy and every data point carries some important information.

Our second crucial insight is that the natural selection of the most critical data points and most relevant core tokens for each data point should go hand in hand. We have demonstrated this through

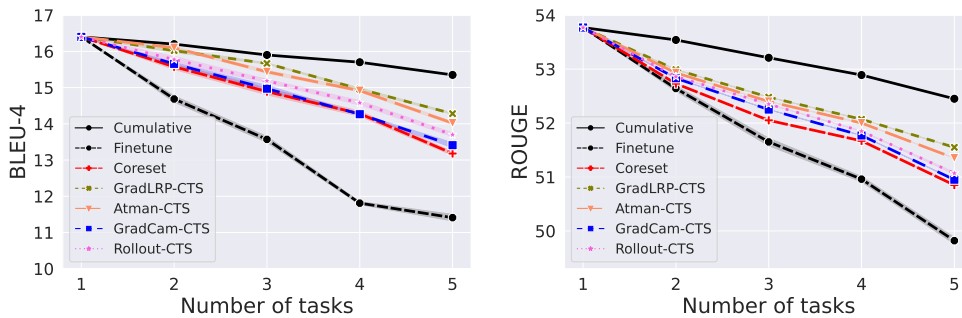

Figure 7: Incremental learning on image caption generation tasks. The left and right panels, respectively, report the BLEU-4 and ROUGE scores for different memory construction methods based on 10% of the data as we finetune BLIP on five mutually exclusive MSCOCO tasks.

the high effectiveness and respective memory size gain in our core tokensets. Here, we hypothesize that the large improvement in performance vs memory size obtained is a result of the cascaded nature of selection. For example, consider a scenario where we are allowed a memory equivalent to 1% of the data. For a traditional core set, this implies discarding 99 % of all data instances. However, for a core tokenset, we can actually extract 10 % of the most meaningful data points and store only 10 % of their most relevant tokens (assuming all data points have equivalent dimensions, as is typically the case in benchmark data sets). If this respective 10 % carries the most relevant information, it is clear why such a 1 % sized core tokensets performs comparably to a 10 % coreset. This fact has consistently been observed throughout the various experiments in our paper.

Finally, this leads us to discuss the limitations of our work and its prospective future. The first limitation is shared with the majority of coreset methods. Infact in its present form, the data retention rate (or memory buffer size) must be chosen in advance unless an additional optimization problem is to be involved. Second, we have deliberately chosen to treat the identification of essential data points and their most informative tokens as two chained steps. This choice was motivated by our desire to show the generality of the core tokenset principle and, respectively, to benchmark a plethora of conceivable selection strategies. It could thus be shown that all core tokenset strategies were beneficial in direct comparison to traditional core sets. However, some of the investigated strategies are inherently amenable to direct combination into a single selection algorithm. In fact, our empirically best-performing combination of GradMatch + GradLRP both rely on the gradient signal. As such, it is conceivable that a joint gradient-based selection strategy can be formulated. Although the computation cost of an additional backward pass is manageable, this would reduce two computations into a single step. More importantly, it opens up the avenue for future work to derive core tokenset guarantees and bounds for epsilon. Related to this, core tokens also share similarities with recent works on token pruning and merging (Meng et al., 2022; Tang et al., 2022; Wei et al., 2023). These methods utilize the notion of token importance to remove unnecessary overhead and improve efficiency during training and inference, but could prospectively also be leveraged to build core tokensets for sequential learning. We view these aspects as a promising future direction, along with further experimentation in other domains.

## 7 CONCLUSION

We have introduced core tokensets for data summarization tailored to the unique architectural properties of transformers. Building on the notion of coresets, we have demonstrated that each data sample can be sufficiently summarized with only a subset of core tokens. Consequently, we have seen the applicability of feature attribution scores in determining the relevance of each input token on the target task. Our empirical investigation in three distinct scenarios has demonstrated that core tokens are a cost-effective way to effectively preserve the stability of a sequentially trained model. About future work, we envision the development of single-stage core tokenset selection strategies, where the relevant set of samples and their informative tokens are identified concurrently.

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

# A APPENDIX

## A.1 DETAILS OF ATTRIBUTION STRATEGIES

In this section, we detail the different selection strategies that ultimately influence the informativeness of the core tokensets and lay out the mathematical details behind each strategy.

**GradCam:** Inspired by the idea proposed in the context of convolutional networks (Selvaraju et al., 2017), we implement a gradient-based feature attribution map for transformers. In particular, we only consider the last layer of the transformer right before the classification head to compute the gradients of the attention heads. The corresponding feature map will obtain a shape of $R^{T \times D}$ where T denotes the number of the tokens and D is the embedding dimension. As a note, GradCam is purely a class-specific approach, where we only consider the gradients concerning the [CLS] token and set the rest of the elements in the first dimension to zero.

**Rollout:** Unlike Gradcam, this strategy is purely attention-based ($A$). The feature relevance map S is usually determined by multiplying the attention maps of all the blocks ($b \in B$) across the transformer layers (Abnar & Zuidema, 2022). It is thereby:

$$S = A^{(1)} \cdot A^{(2)} \cdot A^{(3)} \cdot A^{(B)} \tag{6}$$

**GradLRP:** GradLRP is the second gradient-based approach that amends some of GradCam's limitations. As the name suggests, it combines the gradients of the individual attention maps with the layer-wise propagation score calculated from the target token to the input sample. Given a transformer model with B blocks, we first calculate the attention map ($A^b$) for each block ($b \in B$) comprising h attention heads. We then consider the Hadamard product of the gradients of the attention map ($\nabla A^b$) and the layer-wise relevance score ($L_R$) (Bach et al., 2015) concerning a target class. The final attribution map $\bar{A}^b$ is achieved by calculating the mean ($E_h$) of the product across h attention heads in addition to an identity matrix to account for skip connections in transformers. The final influence matrix $S_{T \times T}$ for a set of tokens T is then defined by multiplying all the attention head scores, where each row consists of the relevance score of each token compared to the other token.

$$\bar{A}^b = I + E_h(\nabla A^b \odot L_R); \quad S_{(T \times T)} = \bar{A}^{(1)} \cdot \bar{A}^{(2)} \cdot \bar{A}^{(3)} \cdot \bar{A}^{(B)} \tag{7}$$

**Atman:** This approach follows a different route, where the relevance of a token is determined by leveraging the perturbation technique. In particular, we perturb the pre-softmax attention score $H$ ($\epsilon R^{h \times t \times d}$) for each token by a factor $f$ and determine their influence over the final target loss. A subtle advantage over its gradient counterpart is that perturbation-based approaches only require forward passes without calculating or storing gradients. Across all heads h and the transformer blocks, we can compute the modified pre-softmax attention scores $H_h$ as:

$$\tilde{H_{h,*,*}} = H_{h,*,*} \odot ((\mathbf{1} - f) + f(\mathbf{1} - f_{k,*}^t)) \ , where \ f_{k,*}^t = \begin{cases} s_{t,k} & if \ \eta \leq s_{t,k} \leq 1 \\ 0 & \text{otherwise,} \end{cases} \tag{8}$$

Here, $\mathbf{1}$ denotes the matrix containing only ones $[1]^{t \times t}$ whereas the $s_{t,k}$ denotes the cosine similarity matrix for the T tokens. Consistent across all the heads, we manipulate the attention scores of token t by a factor of f. Due to the nature of the image sample and the correlation between several patches, we also manipulate the scores of the tokens, which are correlated with the token t. For that, we compute the similarity matrix consisting of the similarity scores for each token compared to other tokens and consequently select column t to perturb individual tokens. We compute the difference between the loss function with the perturbed token and the unchanged token to determine the token relevance.

The highest $I$ score contributes to the largest deviation in the loss function, highlighting the most relevant token.

$$I_t^{target} \approx \mathcal{L}^{target}(z, \theta_{-z_t}) - \mathcal{L}^{target}(z, \theta) \tag{9}$$

Here, $\theta$ denotes the model parameters, whereas the $\theta_{-z_t}$ gives the weight parameters with the perturbed token. Ultimately, we consider the row of the most relevant token from the cosine similarity matrix to construct our core tokenset.

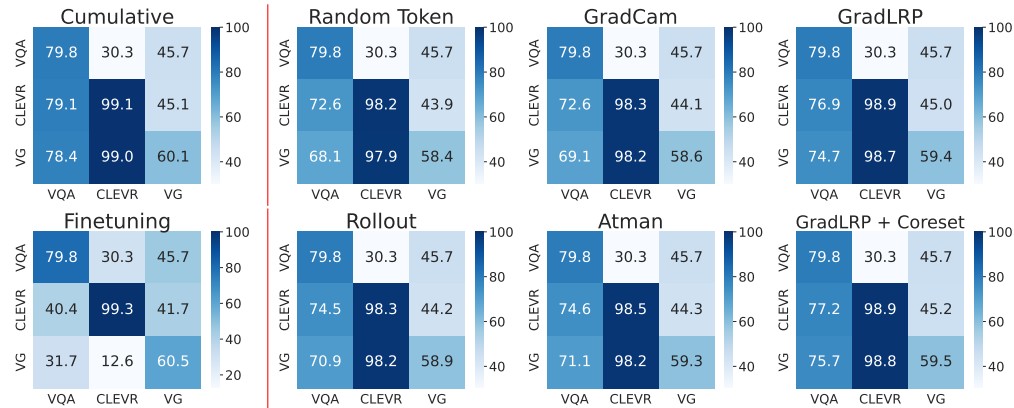

Figure 8: Incremental learning for VQA. Matrices show accuracy (%) of a pre-trained Image-to-Text model (BLIP (Li et al., 2022)) trained sequentially on three datasets VQA-v2 (Antol et al., 2015), CLEVR (Johnson et al., 2017), and VG (Krishna et al., 2017). Experiments are conducted with a 50% retention rate.

## A.2 TRAINING AND DATASET DETAILS

**Sequential image classification:** In this particular task, we have initialized our image transformer with a Vit-B/16 (Wightman, 2019) model pretrained on ImageNet21k. To mimic a sequential learning task, we created a sequence of 5 sub-tasks, each consisting of instances from 20 random ImageNet classes. We then fine-tune our models on each sub-task for 20 epochs with a mini-batch size of 512, using SGD as an optimizer with a learning rate of 0.01 and cosine scheduler. We report the model's accuracy on the current task and the previously encountered sub-tasks at the end of each sub-task.

**Sequential multimodal training:** We have empirically evaluated our proposed approach on two distinct tasks: visual-question answering and image captioning. In the context of this work, we have explored the base model of BLIP, with VIT-B/16 being the image encoder (Li et al., 2022) pre-trained on 14M images, including two human-annotated datasets (COCO and Visual Genome) and three web datasets (Conceptual Captions, Conceptual 12M, SBU captions). We trained on 8 Nvidia-A100 GPU nodes.

*Visual-question answering (VQA):* To formulate a sequential learning task, we incrementally fine-tune our pre-trained BLIP in the order: VQA-v2 (Antol et al., 2015) → CLEVR (Johnson et al., 2017) → VG (Antol et al., 2015). Each sub-task is trained for 20 epochs with a mini-batch size of 32 and Adam as our optimizer with a weight decay of 0.05. The learning rate is warmed up to 3e-4. Additionally, we have considered input images with random crops at a resolution of $480 \times 480$. After each sub-task, we populate our memory using one of the core token selection strategies. We treat BLIP as a generative model to predict answers for a given input image and its corresponding question. Finally, we compare the predicted and target answers and report the scores as our evaluation metric.

*Image captioning:* In the case of an incremental image captioning task, we have segmented MSCOCO into a sequence of five sub-tasks based on the object class types (book, backpack, car, truck, and bottle). Much like the VQA task, we have fine-tuned the model for 20 epochs with Adam as the optimizer. Here, we have considered random crops of images with a resolution of $384 \times 384$.

## A.3 ADDITIONAL RESULTS

### A.3.1 VISUAL QUESTION ANSWERING

To complement the main body, we report the plasticity and stability of the vision-language model BLIP while being trained sequentially on our VQA task, as reported in Fig. 8. For each scenario, a task versus task confusion matrix is shown. The diagonal from the top left to the lower right represents the evaluation performance on a task when it is trained. Respectively, the upper right-hand triangle shows forward transfer, i.e., when the model is evaluated on an unseen task after being trained

on a specific prior dataset. Finally, the lower left triangle quantifies backward transfer, where the model is re-evaluated on an older dataset after being trained on a new dataset.

Not surprisingly, all methods show similarly limited generalization capabilities to unseen tasks, as evident from the upper right triangle. Intuitively, any technique to store a memory of the past, whether core sets or core tokensets, does not affect this generalization capability. The observation primarily underpins that the chosen tasks and pre-trained model are meaningfully picked for continual learning.

The lower triangle is most relevant to evaluating core tokenset selection, as it indicates forgetting between tasks. Here, cumulative training serves as an upper bound. On the contrary, finetuning without replay suffers from catastrophic forgetting, thus marking the lower bound. We can observe that all feature attribution methods improve upon random core token selection. However, a well-picked selection strategy significantly reduces forgetting. For example, when using our core tokenset based on GradLRP, we achieve over 75% accuracy on VQA after three tasks. This is roughly 95 % of the initially achieved accuracy when training the task (79.8 %). In direct comparison, GradCam ends up at 69 %, barely more than the performance of random selection at 68%.

### A.3.2 SEQUENTIAL IMAGE CLASSIFICATION

To complement the main body, we further show the evolution across tasks of different feature attribution techniques for core token selection: GradLRP, Atman, GradCam, and Rollout. The retention rate of the tokens is fixed at 40%. In Fig. 9, the line in black represents the model's cumulative training performance, achieving an accuracy of 85.4%. Upon storing random tokens, we observe a visible decline in performance, with the model only achieving 75.5%. Using feature attribution approaches in determining the relevance of tokens certainly aids the model in preserving their old information. Here, identifying core tokens with the help of Atman has delivered the best results. However, a gradient-based approach such as Grad-LRP only offers a marginal performance degradation of 1.8% in accuracy compared to the reported accuracy of 81.87% for Atman. The baseline approaches in the form of GradCam and Rollout are observed to have a gradual decline in performance with a deviation of 6% and 4% in accuracy, respectively. Ultimately, all techniques improve upon random selection.

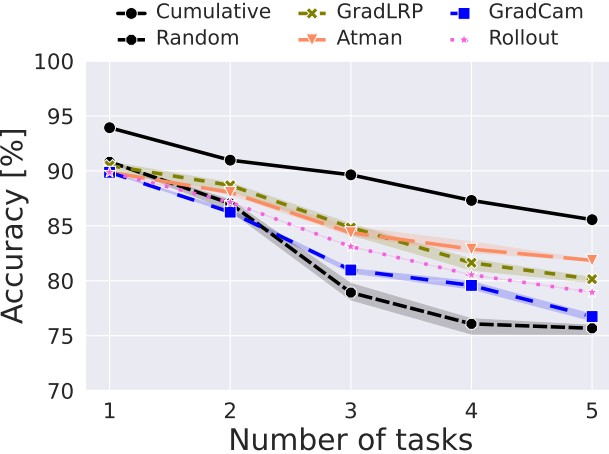

Figure 9: Comparison of core token selection techniques for sequential image classification. Methods vary through the feature attribution technique, yet all demonstrate improvements over a random selection-based core set. In this case, ATMAN is the best approach to identifying core tokens.

### A.4 SEQUENTIAL IMAGE CLASSIFICATION ON SKILL-102 DATASET

We further extend our experimental evaluation to a more challenging dataset with a large variance. To this end, we have chosen a pool of 9 tasks from the continual learning benchmark dataset *SKILL-102*. We have adopted a similar training scenario as in (Ge et al., 2023a) where we consider each dataset, such as *OfficeHome_Clipart, Facial Expressions, Apparel Images*, as a separate task (Ge et al., 2023b).

| Task ID | Task Name | Num Classes | Train Images | Test Images | Ours | CL baselines | | | | | | | | | |
|---|---|---|---|---|---|---|---|---|---|---|---|---|---|---|---|
| | | | | | CTS | CLR | CCLL | EFT | ER | EWC | LWF | ONLINE EWC | PSP | SGD | SGD-LL | SUPSUP |
| 0 | OfficeHome_Clipart | 65 | 3307 | 408 | 87.11% | 85.54% | 79.41% | 58.09% | 41.18% | 1.47% | 0.98% | 1.47% | 1.23% | 1.96% | 0.74% | 45.83% |
| 1 | OfficeHome_Product | 65 | 3361 | 421 | 92.50% | 92.16% | 85.75% | 61.28% | 47.27% | 1.66% | 0.95% | 1.66% | 1.90% | 0.95% | 1.90% | 42.04% |
| 2 | boat_types_recognition | 9 | 1054 | 129 | 91.00% | 89.15% | 79.07% | 60.47% | 100% | 22.22% | 27.7% | 16.67% | 72.22% | 20.93% | 33.33% | 100% |
| 3 | Vegetable_images | 15 | 2565 | 255 | 99.40% | 99.61% | 98.82% | 95.29% | 62.11% | 6.64% | 3.91% | 13.67% | 55.86% | 9.80% | 22.66% | 81.25% |
| 4 | Rice_Image | 5 | 2560 | 255 | 100% | 100% | 100% | 100% | 83.53% | 9.02% | 21.2% | 18.82% | 21.18% | 12.94% | 10.20% | 92.55% |
| 5 | Garbage_classification | 12 | 2556 | 252 | 94.00% | 93.65% | 89.68% | 76.98% | 18.65% | 7.54% | 8.73% | 7.14% | 17.06% | 11.90% | 9.92% | 56.75% |
| 6 | OfficeHome_Art | 65 | 1800 | 250 | 75.43% | 74.80% | 60.40% | 13.60% | 10.80% | 1.20% | 0.80% | 0.80% | 1.20% | 2.40% | 1.60% | 12.80% |
| 7 | Facial_Expressions | 7 | 357 | 49 | 80% | 77.55% | 57.14% | 22.45% | 42.86% | 12.29% | 6.12% | 6.12% | 20.41% | 12.24% | 14.29% | 32.65% |
| 8 | Apparel_Images | 24 | 2482 | 257 | 97.00% | 96.89% | 94.55% | 89.11% | 36.08% | 12.55% | 5.10% | 9.02% | 23.92% | 7.39% | 21.57% | 40.39% |
| | Averaged accuracy | | | | 90.71% | 89.92% | 82.75% | 64.14% | 49.16 | 8.28% | 8.38% | 8.37% | 23.88% | 8.94% | 12.91% | 56.02% |

Figure 10: Extended dataset statistics and observed per-task accuracy after training sequentially on nine sub-tasks of SKILL-102. Core tokensets outperform all other baselines.

To better contextualize these results, we compare core tokensets (CTS) against multiple baseline approaches (Ge et al., 2023a; Kirkpatrick et al., 2017; Cheung et al., 2019; Wortsman et al., 2020; Verma et al., 2021; Singh et al., 2020; Li & Hoiem, 2017; Riemer et al., 2018). Fig. 10, also includes details about each sub-dataset, i.e., number of class labels, training, and test instances. We crafted our core tokensets using GradMatch and AtMan selection strategies at a retention rate of 50%, following insights obtained in the main body's experimentation. We sequentially train each baseline and core tokensets with the datasets in a similar order from OfficeHome_Clipart to Apparel_Images and report the averaged accuracy and the per-task performance at the end of the training. The results in Fig. 10 demonstrate strong performance of core tokensets. Our approach even slightly outperforms the best-performing baseline CLR.

## A.5    LATENCY CONSTRAINT OF CORE TOKENSETS

We now discuss the latency for our core tokensets and single-stage selection strategies, such as coreset and core tokens. The latency incurred by core tokensets is primarily determined by the different core token strategies, considering that the computation of coresets remains consistent, irrespective of the choice of the subsequent subset selection strategy. Considering that we used only gradient-based coresets, it will incur a time complexity of $\mathcal{O}(D \times (FP + BP))$ where D represents the dataset, FP is a forward pass, and BP is a backward pass through the neural network. Meanwhile, core tokens such as GradLRP-based incur a complexity of $\mathcal{O}(D \times H(FP + BP))$ where H is the number of the attention heads. In contrast, Atman does not use backward passes at all, but iterates over all N tokens. It thus has a complexity of $\mathcal{O}(D \times N(FP))$. Consequently, for core tokensets, employing a GradMatch and Atman-based core tokenset will have a time complexity of $\mathcal{O}(D(FP + BP) + C(N \times FP))$. Here, C is the subset selected by the coreset approach, as only this set is relevant for the determination of respective token relevance. The combination with other methods follows analogous reasoning.

## A.6    FURTHER VISUALIZED EXAMPLES OF EXTRACTED TOKEN RELEVANCE WITH DIFFERENT CORE TOKEN STRATEGIES

In this subsection, we visualize the token relevance map computed using different feature attribution techniques to select the *core tokens*. To provide better intuition, we gradually reduce the retention rate from 60% to 20% and visualize which of the *core token* selection strategies could preserve the target object in consideration. We show qualitative examples in Fig. 11- Fig. 13. Referring to Fig. **??**, we observe that random selection, as speculated, fails to capture the relevant information within a sample with the gradual decline in the retention rate. Out of all feature attribution techniques, we notice that GradLRP successfully identifies the target object in consideration, in this case, the elephant, even at a lower retention rate of 20%. Perturbation techniques perform well in preserving the core information and capturing a few background patches due to the closely related patches in image samples. In the

case of Rollout, the closest to the GradLRP approach, we observe a similar trend in establishing a token relevance map. Lastly, Gradcam seems to offer only a marginal improvement over random selection.

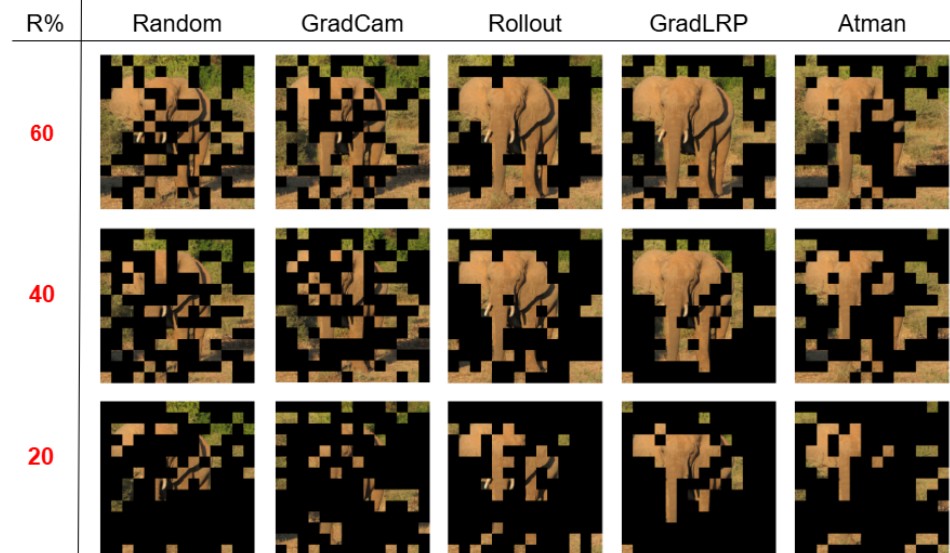

Figure 11: Visualization of the token relevance map constructed with the help of different core token strategies. Object class: *elephant*.

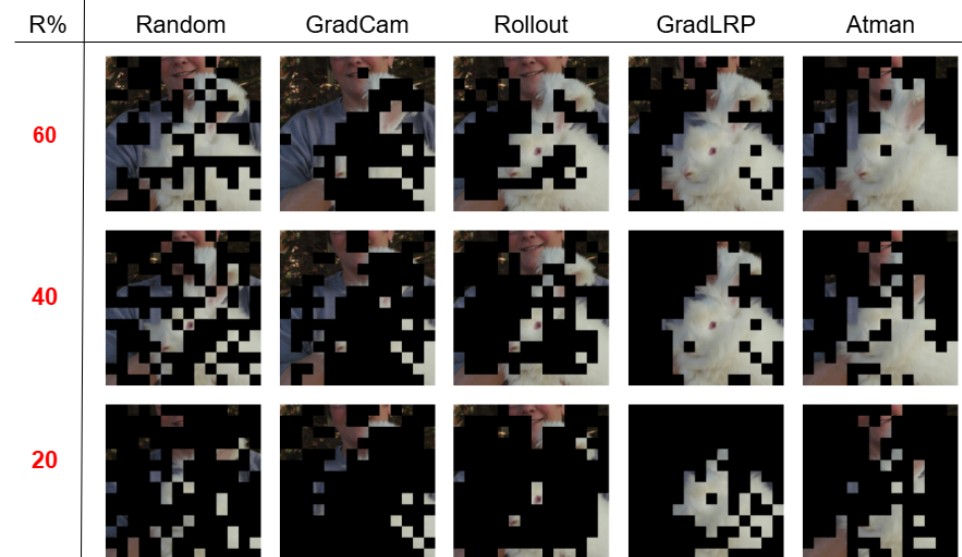

Figure 12: Visualization of the token relevance map constructed with the help of different core token strategies. Object class: *rabbit*.

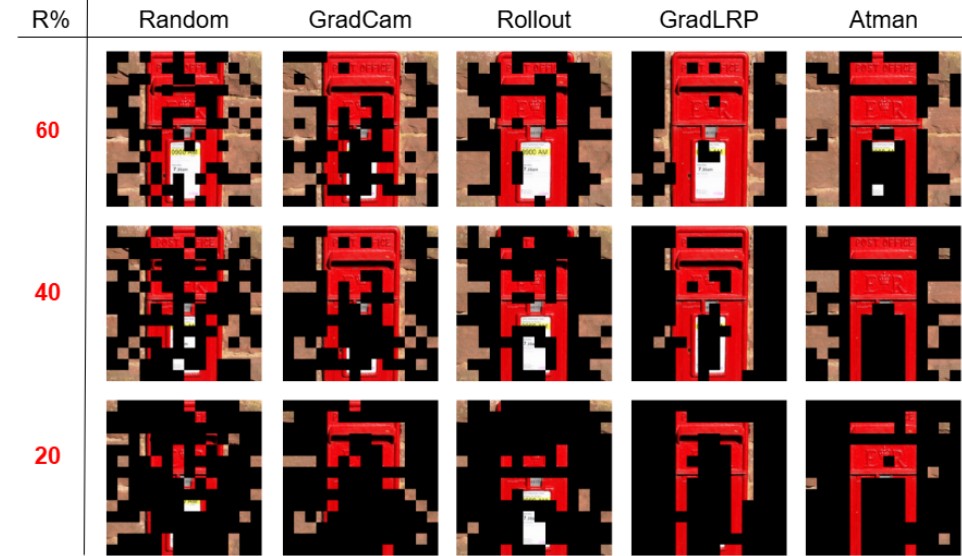

Figure 13: Visualization of the token relevance map constructed with the help of different core token strategies. Object class: *postbox*.

