# OpenReview forum: "Core Tokensets for Data-efficient Sequential Training of Transformers"
_ICLR.cc/2025/Conference — Submitted to ICLR 2025_

### Official Review · Reviewer_ey6M · 2024-11-03

**Soundness:** 4
**Presentation:** 4
**Contribution:** 3
**Rating:** 5
**Confidence:** 4

**Summary:**

This paper introduces a novel concept of core tokensets, which selects only the most informative / important tokens from training samples as replay data for continual learning.
Previous works also study how to select more important samples for memory efficiency, but they do not operate at the token level.
This work further enhances this idea with a two-step selection process. First, they select important training samples with coreset selection methods (like GradMatch). Then, they identify and retain only the most relevant tokens from those selected samples using methods like Grad-Cam, Grad-LRP, and Atman.
To make transformers amenable to such partial inputs, this paper also employs random token dropping during training.
In summary, core tokensets provide an advanced way to sub-select training data at the token level, leveraging the properties of transformer architectures to enable highly efficient data replay for continual learning.

**Strengths:**

- The paper introduces the concept of core tokensets, that selects the most informative tokens from important data samples.
- The authors extensively evaluate their approach across multiple sequential learning tasks, including image classification, visual question answering, and image captioning.
- The paper explores various strategies for selecting important tokens, including Grad-Cam, Rollout, GradLRP, and Atman, providing a thorough comparison.
- I like this paper's straightforward writing style, as it provides detailed explanations of existing methods and comprehensive results, rather than over-emphasizing a supposedly novel method.

**Weaknesses:**

- The proposed token-based core tokensets are specifically designed for transformer models, which are widely used today. But this idea may have limited applicability to inspire other fields.
- Although this paper presents detailed results, I think its main drawback is the lack of theoretical findings, which makes the work silghtly looks like an experimental report. As shown in Table 1  and Table 2, the best results for Core Tokenset are obtained by combining the top Coreset selection and Core Token selection methods. However, the performance differences between all these approaches are small. This gives the impression that the paper simply stating, "A is good, B is good, so A+B is better." However, how to know which one is better? To improve the paper, I suggest conducting more theoretical studies to explain the selection process.

**Questions:**

What is the theoretical bounds or guarantees on the performance of core tokensets?

---

> ### Author Response · Authors · 2024-11-22
> **Response to review and summary of changes in the revised paper upload**
>
> We sincerely thank the reviewer for the detailed assessment of our work. In the following we address each stated weakness and raised questions individually.
>
> ***W1) The proposed token-based core tokensets are specifically designed for transformer models, which are widely used today***
>
> We wish to emphasize that coresets, as an architecture agnostic concept, have been widely studied in countless works over the last decades. We motivate and introduce these carefully in our methods section and later compare them to two SOTA techniques.
> As the reviewer stated, transformers are widely used today and have become the de facto standard architecture for the majority of computer vision and natural language processing tasks. Consequently, we argue that a memory buffer technique specifically catering to the transformer architecture is not necessarily a weakness of our work, but a testament to the current state of the field. Conversely, core tokensets are the first method leveraging the presently predominant architecture in modern machine learning and we show that we can gain significant additional advantages over the traditional coreset approach here.
>
>
> ***W2) the performance differences between all these approaches are small. This gives the impression that the paper simply stating, "A is good, B is good, so A+B is better." However, how to know which one is better? To improve the paper, I suggest conducting more theoretical studies to explain the selection process.***
>
> We certainly agree with the reviewer that additional theoretical studies are always beneficial to the readers. We have also discussed this point transparently in the discussion section with respect to limitations and future work of our main body. Pertaining to the fact that our core tokenset idea is quite unique in itself, we thought that it would be more meaningful to introduce the main idea at first and corroborate it as widely as possible empirically first (much like many novel ideas, such as Dropout, in the past).
> Moreover, if we recall the results from our work’s extensive investigation, the best performing core tokensets often combine GradLRP based core tokens with GradMatch based coreset.  Both being based on gradients, we believe that our empirical investigation paves the path for future theoretical guarantees to be derived, given that the Grad-Match algorithm already has theoretical guarantees for core sets.
>
> However, one critical point to consider is that the combination not only improves the performance in terms of accuracy by a certain margin, but also touches a key constraint in incremental learning, i.e. the size of the memory buffer. As highlighted in the experimental and discussion section, core tokensets essentially achieve a similar level of performance compared to core tokens or coresets, but with significantly less amount of data in the replay buffer, in some cases only half or even ten times less samples.
>
>
> ***Q1) What is the theoretical bounds or guarantees on the performance of core tokensets?***
>
> As described transparently in our extensive discussion section, one limitation of our work is that we do not presently derive theoretical bounds. As noted in the above response to W2, this is a deliberate choice to allow investigation of more complex combinations of popular strategies. We note that GradMatch, as a core set method, has well-known theoretical guarantees and posit that its combination with GradLRP (i.e. a purely gradient based core tokenset approach) would allow to derive respective guarantees as well. We did presently not investigate this avenue further, both due to content/space constraints, and because it is not clear that all combinations would allow for bounds to be derived. For instance, Atman relies on perturbation through forward passes rather than gradients and would require a fundamentally different approach to deriving bounds, yet performs comparably to GradLRP in our core tokenset investigation. We thought that this insight is very valuable for readers. Now that our extensive empirical investigation has shed light on which algorithms are of practical value, we believe that the path has been paved for future work to investigate the complementary theoretical side.

---

> ### Author Response · Authors · 2024-11-25
> **Kind Reminder: Follow-up on Rebuttal Feedback**
>
> Thank you for your thoughtful and constructive feedback on our submission. We have carefully addressed your concerns and provided detailed responses in our rebuttal. As the deadline is fast approaching, we kindly remind you to review our rebuttal at your earliest convenience. If there are any remaining issues or further clarifications needed, we would be more than happy to address them.

---

### Official Review · Reviewer_uaoE · 2024-11-04

**Soundness:** 3
**Presentation:** 2
**Contribution:** 2
**Rating:** 5
**Confidence:** 4

**Summary:**

The paper proposes a new way to extract core tokenset. Overall, the paper is well written. My comments are as below:
1. I can see that the tasks used in the paper are generally easy task with 90%+ accuracies. I'm wondering the performance of the proposed method on more difficult tasks.
2. I want to see more visualized results. Will that be possible to show the extracted token, and corresponding image patches?
3. I think the work can be related to model explainability, will the authors cite some related works and explain the model?
4. How to define the relevancy among tokens? I still think the proposed method lack theoretical basis; also, I want to see more model training details.

**Strengths:**

The proposed method is with practical value, which could be applied to lots of applications.

**Weaknesses:**

As can be found in the summary.

**Questions:**

I want to see how the authors define the "discriminative tokens". More mathematical reasoning should be provided.

---

> ### Author Response · Authors · 2024-11-22
> **Response to review and summary of changes in the revised paper upload**
>
> We thank the reviewer for their feedback. Subsequently, we address each of the reviewers comments.
>
> ***W1) I can see that the tasks used in the paper are generally easy tasks with 90%+ accuracies. I'm wondering the performance of the proposed method on more difficult tasks***
>
> Please note that the applicability of our method does not necessarily depend on the nature of the underlying task. In general, the performance of core tokensets will be bounded by the performance achievable through joint training, as are all continual learning methods that keep a subset of the data for future training.
>
> To nevertheless accomodate the reviewers concern, we added an experiment on such a challenging task, i.e. ImageNet-R to Section 5.3 of the paper. In this setting the joint baseline only reaches 81% accuracy. Our core tokensets approaches outperforms all other continual learning methods, including recent SOTA approaches like InfLoRa.
>
> ***W2) I want to see more visualized results. Will that be possible to show the extracted token, and corresponding image patches?***
>
> In addition to the visualizations already included in the Appendix, we added further examples in our revised manuscript. These visualizations highlight coretoken selection at different retention rates and demonstrate qualitative selection differences across selection techniques.
>
>
> ***W3) I think the work can be related to model explainability, will the authors cite some related works and explain the model? How to define the relevancy among tokens? I want to see more model training details***
>
> 3.1 **Explainablity works:** We agree that his work is closely related to model explainability, since we heavily rely on respective explainability methods for core token selection. Consequently, our discussion in Section 3 contains multiple citations to relevant methods. We revised our prior formulation of “interpretability” methods to “explainability” to highlight this connection
>
> 3.2 **Relevancy Definition:** As discussed in Section 3, we derive relevancy among tokens from the attribution maps of the respective explainability methods. We have already provided detailed descriptions and mathematical equations for the relevancy calculation of each employed method in Appendix A.1 in our initial submission and politely point to them.
>
> 3.3 **Model training details:** Due to the page limit constraints, Section 4 only contains the key aspects of our model training details. We have provided further details for each task in Appendix A.2 in our original submission.
>
> ***Q1) I want to see how the authors define the "discriminative tokens". More mathematical reasoning should be provided.***
>
> Similar to prior works, we define a token as “discriminative” if it allows us to deduce the correct output (e.g. the correct label in classification). We note that our methods draw inspiration from explainability methodology, see above reply to weaknesses, and describe the math behind token extraction in appendix section A.1 for all investigated algorithms.

---

> > ### Comment · Reviewer_uaoE · 2024-12-03
> > **Not satisfied with the results**
> >
> > I believe the authors have partially addressed my issue. However, I'm not entirely satisfied with the results for W1 as outlined in Section 5.3. Additionally, for Q1, there is still a lack of a specific algorithm. Therefore, I would increase my rating to 5, but I wouldn't prefer a higher rating.

---

> ### Author Response · Authors · 2024-11-25
> **Kind Reminder: Follow-up on Rebuttal Feedback**
>
> Thank you for your thoughtful and constructive feedback on our submission. We have carefully addressed your concerns and provided detailed responses in our rebuttal. As the deadline is fast approaching, we kindly remind you to review our rebuttal at your earliest convenience. If there are any remaining issues or further clarifications needed, we would be more than happy to address them.

---

> ### Author Response · Authors · 2024-12-03
> **Response to the Reviewer uaoE**
>
> We thank the reviewer for their feedback and for raising the score.
>
> We would also like to draw the reviewer's attention to the Appendix in Section A.4, where we added another experiment on the SKILL-102 dataset, a challenging benchmark dataset for incremental learning (https://paperswithcode.com/dataset/skill-102). We believe that we have extensively evaluated the robustness of the *core tokensets* across a wide variety of tasks with different complexities, such as classification (standard ImageNet, a more challenging variant Imagenet-R, and SKILL-102 dataset), and further scaled it to multimodal tasks in the form of visual question answering and image captioning that standard CL baselines such as ER didn't evaluate on. Each sub-task of incremental multimodal tasks is unique yet challenging, with the SOTA approaches on image captioning only reporting a BLEU-4 score of 44. In section 5.3 and section A.4, we also evaluate core tokensets on two standard continual learning (CL) benchmark datasets used in SOTA CL approaches as suggested by the reviewer L2dW and later acknowledged the findings. However, it is essential to consider that our approach lacks any SOTA method, given the novelty of this work and the fact that the baselines we compared are fundamentally different. Yet, from Table 2 in section 5.3, it is evident that core tokensets also stand competitive against standard CL baselines.
>
> As an additional note, the objective of our work is not to beat the baseline single-task performance but to reduce the performance gap ***(catastrophic forgetting)*** when the models are trained sequentially on a set of tasks. We would appreciate it if the reviewer could point us toward an even more challenging incremental learning setup.
>
> We could definitely add a moral formal algorithm for our approach in the camera-ready version. Again, we thank the reviewer for the time and effort and hope the reviewer reconsiders their evaluation.

---

### Official Review · Reviewer_JrZt · 2024-11-04

**Soundness:** 3
**Presentation:** 3
**Contribution:** 2
**Rating:** 6
**Confidence:** 3

**Summary:**

The paper introduces the concept of core tokensets for data-efficient sequential training of transformers, demonstrating improved performance and reduced memory usage in various vision and language tasks, including image classification, visual question answering, and image captioning.

**Strengths:**

- **Novel Concept**: The concept of core tokensets, which select the most informative tokens within informative data points, represents a novel and efficient way to summarize datasets for sequential training of transformers.

- **Solid Performance**: Core tokensets demonstrate comparable effectiveness to other data selection methods (e.g., Coreset, Core Token) while substantially reducing memory usage.

- **Versatility**: Core tokensets was shown effective on multiple different vision and language tasks inlcuding image classification, visual question answering, and image captioning

- **Presentation**: The paper is clearly written and easy to follow.

**Weaknesses:**

While core tokensets significantly reduce memory usage, their two-step approach—first selecting informative data points and then identifying key tokens within those points—may introduce more latency in dataset summarization compared to single-step methods (e.g., Coreset, Core Token).

**Questions:**

- Have you compared the latency of core tokensets with other data selection methods?

---

> ### Author Response · Authors · 2024-11-22
> **Response to review and summary of changes in the revised paper upload**
>
> We sincerely thank the reviewer for their time and effort in providing us with feedback.
>
> ***W1) While core tokensets significantly reduce memory usage, their two-step approach—first selecting informative data points and then identifying key tokens within those points—may introduce more latency in dataset summarization compared to single-step methods (e.g., coreset, core token)***
>
> We agree with the reviewer that the paper would benefit from a more detailed study on latency and complexity of different approaches. Consequently, we added a corresponding section to main paper (Section A.5).
>
> In general, we would first like to highlight that we have made a deliberate choice to pursue a two-step strategy, as discussed transparently in our discussion section. The primary two reasons are that a) the core token selection and investigation itself is novel and b) that a two-step approach allows for a much more nuanced investigation of algorithmic couplings. Not all methods would be directly combinable into a single stage algorithm, e.g. gradmatch and gradLRP would both leverage a backward pass to compute gradients, but AtMan is forward-only.
>
> In that sense, selecting a specific approach for core token selection offers novel trade-offs. For example, identifying core tokens with GradLRP requires one forward and backward pass for each sample. This complexity/latency is comparable to a coreset build with gradmatch, but the core tokens approach slightly outperforms the coreset. On the other hand, utilizing a perturbation approach like AtMan increases latency, but scales to large model sizes, as it requires no backward pass. Consequently, the approach agnostic formulation of core tokens offers a more flexible framework that can account for different settings and requirements.
>
> Nonetheless, applying a two-step method will naturally concur an increase in latency. However, it is important to note that the second core token stage only needs to be calculated on the already reduced coreset. For example, considering the example from Table 1 in the paper. We would first construct a coreset at 10% retention rate. Further reducing this memory buffer to a core tokenset thus comes with a latency overhead of roughly 10% (see similar complexity above). However, that 10% overhead results in a further 10x reduction in memory buffer size while maintaining downstream performance.

---

> > ### Author Response · Authors · 2024-11-25
> > **Kind Reminder: Follow-up on Rebuttal Feedback**
> >
> > Thank you for your thoughtful and constructive feedback on our submission. We have carefully addressed your concerns and provided detailed responses in our rebuttal. We kindly remind you to review our rebuttal at your earliest convenience. If there are any remaining issues or further clarifications needed, we would be more than happy to address them.

---

> > > ### Comment · Reviewer_JrZt · 2024-11-26
> > >
> > > Thank you for your response. The authors addressed my concerns, and I will maintain my score.

---

> > > > ### Author Response · Authors · 2024-11-29
> > > > **Response to the Reviewer JrZt**
> > > >
> > > > Thank you for going through our rebuttal and acknowledging our work.

---

### Official Review · Reviewer_Ld2W · 2024-11-05

**Soundness:** 2
**Presentation:** 3
**Contribution:** 2
**Rating:** 6
**Confidence:** 2

**Summary:**

This paper explores the incremental learning task using transformers.  The authors focus on improving the efficacy of replay methods, where they argue for storing key tokens rather than entire images.  They evaluate performance on image classification, VQA, and captioning tasks, where they improve performance over other coreset methods.

**Strengths:**

1. I do not know of another paper that explores the effect of storing a subset of tokens for replay in incremental learning
2. The diversity of tasks in the experiments is good- many incremental learning papers just evaluate on one task
3. Using a subset of tokens could also benefit efficiency of replay methods.

**Weaknesses:**

1. Identifying a subset of tokens that does as well as the whole sample has been explored in prior work.  In particular, there are token pruning/merging methods (e.g., [A,B]) as well as methods used to for faster training (e.g., [C]).  The authors don't discuss or compare to these methods (I only gave examples, but there are many papers on this topic that should be included in the discussion).  As such, one could see this paper as simply a new application of a known technique.

2. The authors do not compare to the state-of-the-art, but rather restrict their comparisons to closely related papers.  As such, it isn't clear how the proposed approach compares.  The authors should consider more recent methods as well as those from the broader literature for their comparisons (e.g., [D,E]).

3. It would be interesting to see how the proposed method works on a more diverse set of datasets.  While the diversity of tasks was appreciated, they are typically limited to a single dataset (and ImageNet is also restricted to a subset of categories.  A more thorough evaluation like those in the related work would improve the paper and provide additional insight into how well the proposed approach works (e..g,, [F]).

[A] DynamicViT: Efficient Vision Transformers with Dynamic Token Sparsification. NeurIPS 2021

[B] Beyond Attentive Tokens: Incorporating Token Importance and Diversity for Efficient Vision Transformers. CVPR 2023

[C] Masked Autoencoders Are Scalable Vision Learners. CVPR 2022

[D] Adaptformer: Adapting vision transformers for scalable visual recognition. NeurIPS 2022

[E] InfLoRA: Interference-Free Low-Rank Adaptation for Continual Learning. CVPR 2024

[F] CLR: Channel-wise Lightweight Reprogramming for Continual Learning. ICCV 2023

**Questions:**

See weaknesses.

---

> ### Author Response · Authors · 2024-11-22
> **Response to review and summary of changes in the revised paper upload**
>
> We thank the reviewer for the detailed assessment of our work and the pointers to improve our manuscript. We subsequently address each of the stated weaknesses/questions individually.
>
> ***W1) There are token pruning/merging methods (e.g., [A,B]) as well as methods used for faster training (e.g., [C]). The authors don't discuss or compare these methods***
>
> We agree with the reviewer that while being a novel method for incremental learning, coretoken selection shares similarities with existing literature on token pruning. Consequently, we added a respective paragraph to the discussion (Section 6).
>
> ***W2) The authors do not compare the state-of-the-art, but rather restrict their comparisons to closely related papers. As such, it isn't clear how the proposed approach compares.***
>
> We note that there technically exists no state-of-the-art (SOTA) for core tokenset subset selection, as we are the ones introducing this novel concept. Because of this, we compared the conceptually closest concept, namely core sets. Here, the methods we compared against, namely GradMatch and CRAIG are indeed the current SOTA.
> However, we also agree with the reviewer that a further comparison against substantially different approaches that are SOTA in sequential/continual learning will help to further contextualize the strengths of core tokensets in the broader literature. To that end, we added an experiment on ImageNet-R in Section 5.3, which included a direct comparison with various SOTA methods. Core tokensets remain competitive in this setting, even outperforming all other approaches, but we carefully note that core tokensets are not mutually exclusive, but complementary to these techniques.
>
> ***W3) It would be interesting to see how the proposed method works on a more diverse set of datasets. While the diversity of tasks was appreciated, they are typically limited to a single dataset (and ImageNet is also restricted to a subset of categories.***
>
> In addition to the now included ImageNet-R, we also conducted further experiments (App. A4) on a more diverse dataset consisting of 9 unique sets, as introduced in SKILL-102. Again, core tokensets outperform all other methods, demonstrating its strength and general utility in more diverse continual learning setups.

---

> > ### Author Response · Authors · 2024-11-25
> > **Kind Reminder: Follow-up on Rebuttal Feedback**
> >
> > Thank you for your thoughtful and constructive feedback on our submission. We have carefully addressed your concerns and provided detailed responses in our rebuttal. We kindly remind you to review our rebuttal at your earliest convenience. If there are any remaining issues or further clarifications needed, we would be more than happy to address them.

---

> > > ### Comment · Reviewer_Ld2W · 2024-11-28
> > >
> > > Thanks, that is helpful, I have increased my score to 6, but did not raise it farther due to my first weakness, that the authors seem to indicate stands.  I don't think this is enough rejecting the paper over, but it does diminish the potential impact.

---

> > > > ### Author Response · Authors · 2024-11-29
> > > > **Response to Reviewer Ld2W**
> > > >
> > > > Thank you for going through our rebuttal and acknowledging our work.

---

### Author Response · Authors · 2024-11-22
**Uploaded a revised pdf based on reviewers' feedback**

We thank all reviewers for the thorough feedback. We appreciate that the reviewers found our work novel (Ld2W, JrZt, uaoE), well-written (JrZt, uaoE, ey6M) and extensively evaluated (Ld2W, JrZt, ey6M).

As noted in the individual responses, we have carefully taken all comments into account and updated our manuscript accordingly. We have made use of the additional 10th page to include the reviewers’ primary suggestions and maintain legibility. Further parts are added to the appendix. The respective changes are all highlighted in yellow in the uploaded revised PDF.

In short, we made the following adjustments:
1. Experimental evaluation on a new dataset ImageNet-R, a challenging dataset used to benchmark recent works on incremental learning. Coretokens achieve competitive scores, even outperforming all other approaches, demonstrating its general utility. We emphasize that core tokensets are nevertheless complementary to these methods. Please see main body Sec 5.3

2. Additional experiment on a more diverse dataset consisting of 9 unique tasks from SKILL-102, a benchmark incremental learning dataset. Again, coretokens outperform all other baseline approaches. Please see appendix A.4
3. Sketched the computational aspect of different core token strategies. Please see appendix A5
4. Additional qualitative examples that visualize core token selection with different selection strategies. Please see figure 2 in the main body and additional examples in appendix A6.
5. Included references to token pruning in light of future work in the discussion section. Please see main body Sec. 6

---

### Meta-Review · Area_Chair_xkCg · 2024-12-17

**Metareview:**

This paper studies the identification of a subset of tokens that can maintain the performance of vision transformers for tasks such as image classification, visual question answering, and image captioning. The reviewers acknowledged the strengths of the paper, including its novelty and the diverse verification across three distinct tasks. However, they also highlighted several significant weaknesses.

Following the rebuttal, several concerns remain:

* Comparison and relationship to existing works: While the authors provided a brief discussion, it did not persuade the reviewers.
* Evaluation on more challenging tasks: The authors’ response did not sufficiently address this concern and failed to convince the reviewers.
* Inadequate performance: Although results on another dataset were included, the overall performance remains unsatisfactory.
* Furthermore, two reviewers raised concerns regarding the absence of theoretical analysis.

Ultimately, two reviewers lean toward rejecting the paper, following the rebuttal discussion with the authors. Given that some unresolved concerns are significant and remain unaddressed, the AC recommends not accepting this submission at this stage.

**Additional Comments On Reviewer Discussion:**

All reviewers participated in the rebuttal discussion. While two reviewers increased their ratings, the two reviewers who initially gave borderline reject scores (5) were unwilling to raise their ratings further, citing unresolved concerns which are listed in the meta-review.

---

### Decision · Program_Chairs · 2025-01-22

Reject